# Antioxidative Characteristics of Chicken Breast Meat and Blood after Diet Supplementation with Carnosine, L-histidine, and β-alanine

**DOI:** 10.3390/antiox9111093

**Published:** 2020-11-07

**Authors:** Wieslaw Kopec, Dorota Jamroz, Andrzej Wiliczkiewicz, Ewa Biazik, Anna Pudlo, Malgorzata Korzeniowska, Tomasz Hikawczuk, Teresa Skiba

**Affiliations:** 1Department of Functional Food Products Development, Faculty of Biotechnology and Food Sciences, Wrocław University of Environmental and Life Sciences, 37 Chelmonskiego Str., 51-650 Wrocław, Poland; wieslaw.kopec@upwr.edu.pl (W.K.); anna.pudlo@upwr.edu.pl (A.P.); teresa.skiba@upwr.edu.pl (T.S.); 2Department of Animal Nutrition and Feed Management, Faculty of Biology and Animal Husbandry, Wrocław University of Environmental and Life Sciences, 38C Chelmonskiego Str., 51-630 Wrocław, Poland; dorota.jamroz@up.wroc.pl (D.J.); andrzej.wiliczkiewicz@upwr.edu.pl (A.W.); tomasz.hikawczuk@gmail.com (T.H.); 3Department of Agricultural Engineering and Quality Analysis, Wroclaw University of Economics and Business, 53-345 Wrocław, Poland; ewa.biazik@ue.wroc.pl

**Keywords:** histidine dipeptides, antioxidant potential, amino acids, chickens

## Abstract

The objective of the study was to test the effect of diets supplemented with β-alanine, L-histidine, and carnosine on the histidine dipeptide content and the antioxidative status of chicken breast muscles and blood. One-day-old Hubbard Flex male chickens were assigned to five treatments: control diet (C) and control diet supplemented with 0.18% L-histidine (ExpH), 0.3% β-alanine (ExpA), a mix of L-histidine\β-alanine (ExpH+A), and 0.27% carnosine (ExpCar). After 28 days, chicken breast muscles and blood samples were analyzed for the antioxidant enzyme activity (catalase (CAT), glutathione peroxidase (GPx), superoxide dismutase (SOD)), carnosine and anserine content, amino acid profile, and anti-radical activity (ABTS, DPPH, ferric reducing antioxidant power (FRAP)). The results of the study showed that carnosine supplementation effectively increased body weight and breast muscle share in chicken carcasses. Carnosine and L-histidine supplementation with or without β-alanine increased carnosine content in chicken breast muscles up to 20% (*p* = 0.003), but the boost seems to be too low to affect the potential antioxidant capacity and amino acid content. The β-alanine-enriched diet lowered dipeptide concentration in chicken blood serum (*p* = 0.002) and activated catalase in chicken breast muscles in relation to the control group (*p* = 0.003). It can be concluded that histidine or dipeptide supplementation of chicken diets differently affected the total antioxidant potential: in breast muscles, it increased dipeptide content, while in blood cell sediment (rich in erythrocytes), increased SOD and GPx activities were observed.

## 1. Introduction

Carnosine is a naturally occurring dipeptide consisting of β-alanine (β-Ala) and L-histidine (L-His), which is considered to be an immunomodulator. The dipeptide promotes tissue repairing and induces the hypoglycemic effect in several species, including humans [1]. Currently, it is emphasized that an increased content of dipeptides can improve the antioxidant status of muscles and, in addition, meat quality [2]. The possibility of an increase of the histidine dipeptide content in animals’ muscles or other tissues, especially in the brain, by diet supplementation with this amino acid or carnosine has been studied in References [3,4,5,6]. Poultry muscles are one of the richest sources of histidine dipeptides [7]. Thus, intensive studies have been carried out on the possibility to elevate muscle dipeptide content through the use of histidine-rich components in chicken diets [2,8,9,10,11]. Moreover, it is important to point out that histidine, β-alanine, or carnosine supplementation can increase the body weight of the animals [2,12] and thus improve the economical aspects of chicken production.

The dipeptide content in the muscles of vertebrates is much higher than in serum, where its level is controlled by carnosinase (CN1) [13]. Investigations over the last few years showed an increase of carnosine in human muscles after supplementation of their diet with β-alanine [14,15]. Apart from β-alanine, the effects of histidine or histidine dipeptide addition to the animal diet were also studied [16,17]. However, contradictory results concerning the influence of poultry and fish diets supplemented with L-histidine and β-alanine on the dipeptide level, especially anserine in muscles, were obtained. For example, in a short-term experiment after β-alanine supplementation of a poultry diet, an increase in carnosine could be registered. However, the total dipeptides failed to show a significant difference [4]. In fish muscles, β-alanine diet supplementation with or without histidine did not lead to an increase in dipeptide content in muscles [12]. In contrast to that, histidine supplementation in poultry diets resulted in an increase in carnosine concentration in breast muscles [2,18,19]. However, the majority of available studies were focused on the effect of one simple dietary component (i.e., histidine, β-alanine, or carnosine) on the quality and content of histidine dipeptides in poultry tissues.

Since data concerning the effect of carnosine and its components, β-alanine and histidine, on the dipeptide concentration in poultry muscle are controversial, the aim of this study was to test the effect of diet supplementation with β-alanine, L-histidine, and carnosine on: (i) histidine dipeptide content in muscles and blood, (ii) antioxidative status of chicken breast muscle and blood, and (iii) chicken performance.

## 2. Materials and Methods 

### 2.1. Animals, Experimental Design, Diets, Performance Indices

One-day-old Hubbard Flex (Hubbard Co, Pawlow Trzebnicki, Poland) male chickens (*n* = 240) with an average initial body weight of 43 ± 2 g were randomly assigned to five treatments: (i) control (C), (ii) control with added L-histidine (ExpH), (iii) control with added β-alanine (ExpA), (iv) control with both L-histidine and β-alanine (ExpH+A), and (v) control with added carnosine (ExpCar). L-His and β-Ala were purchased from Bulk Powders (Essex, UK), while carnosine from Bulk Supplements (Nevada, USA). Diets were given in a mashed form and contained 215 g/kg of crude protein and 12 MJ metabolizable energy/kg (Table 1). The composition of the control diet used in all treatments was calculated on the basis of previously determined analytical data for all feed components using the simple linear optimization method. The vitamin/mineral premix (composition is described in Table 1) with coccidiostatic Diclazuril (Fipharm Co. Ltd., Haikou City, China) was applied. The control diet contained 5.6 g/kg histidine (36 mM) (Table 2). ExpH was supplemented with a 1.8 g/kg addition (12 mM) of L-histidine, which represents a 30% higher amino acid level than in the control. To the ExpA diet, β-alanine exclusively was added in the amount of 3.2 g/kg (36 mM). In the diet ExpH+A, both L-histidine and β-alanine were incorporated (1.8 g/kg His and 3.2 g/kg β-Ala). Additionally, the effect of carnosine supplementation (2.7 g/kg, 12 mM) was examined in ExpCar. β-alanine was added in the amount equal to 36 mM (3.2 g/kg), which exceeded about 3 times the additions of histidine and carnosine in moles (12 mM), to increase the amount of the dipeptides in the meat. The supplementation level of the diet was designed on the basis of the literature data [4,11,17,19], which showed the effect of histidine addition started from 1.0 g/kg feed, while β-alanine application (if it is used alone) should be higher than histidine. Carnosine incorporation into the diet was equivalent to the amount of L-His in the diets enriched in this amino acid. Each treatment comprised of six replications with 8 chickens kept in battery cages (floor 0.6 m^2^, height 60 cm). The breeding program met the requirements of Council Directive 2007/43/EC [20].

All diets (control and experimental) contained a balanced amount of amino acids expressed as g/kg feed: Asp 21.5; Thr 8.4; Ser 10.7; Glu 47.7; Pro 14.5; Gly 8.8; Ala 9.5; Val 9.8; Ile 8.8; Leu 17.0; Tyr 5.6; Phe 10.7; Lys 13.2; Arg 16.2; Cys 3.4; Met 5.7 and Trp 2.4.

During the whole experiment, chicken body weight (BW) and feed consumption (FC) were controlled, and feed conversion ratio (FCR) was calculated. All chickens were weighed individually using a technical balance (0.1 g). The feed consumption was registered per replication and calculated for the period from 1 to 28 days post-hatch. The period of 28 days of rearing was chosen for the model experiment as this is a time of intensive physiological growth. Twelve randomly selected chickens from each treatment were slaughtered by cervical dislocation according to Council Regulation 2009/1099/EC [21]. All procedures that were carried out with the animals have been approved by the Local Ethics Commission for Experiment on Animals (6/2009 of 19 January 2009 of the 2nd Local Ethical Committee for Animal Experiments in Wroclaw).

### 2.2. Blood Collection and Muscles Preparation for Analyses

Blood was collected to tubes with heparin during slaughtering of the randomly selected 12 chickens per treatment (five treatments, two chickens per cage) at the age of 28 days. The tubes were immediately cooled down in an iced/water bath and centrifuged (5000× g). Then, the yellow plasma layer separated from blood cells was collected and stored at −80 °C until analysis. The activity of antioxidant enzymes, as well as total antioxidant activity indices (ABTS—reducing activity of 2,2-Azinobis- (3-ethylbenzothiazoline-6-sulfonic acid) radicals; DPPH—scavenging activity of 1,1-Diphenyl-2-picrylhydrazyl radical; FRAP—ferric reducing antioxidant power) were determined in blood plasma. The sediment of blood cells used for the enzyme activity assay was washed three times with cold 0.15 M NaCl solution followed by centrifugation at 5000 × g. Protein content in sediment was determined using the biuret method [22]. Wet breast muscles (*pectoralis major* and *pectoralis minor*) (*n* = 60) were excised from the chickens within 10 min of slaughter and blood collection, then chilled in a freezer at −20 °C for 30 min, then transferred to −80 °C and kept until analysis. Muscle tissue was homogenized (Mixer B-400, Buchi, Switzerland) with redistilled iced water (1:1) and centrifuged (Sigma Ltd., Newtown, UK) to obtain the hydrophilic fraction [23]. The fraction was further used for total antioxidant activity and dipeptide content analyses.

### 2.3. Chemical Composition

The chemical composition of feed compounds and complete diets was determined according to the standard methods AOAC (2005): the nitrogen by the Kjeldahl method (984.13) using Kjeltec 2300 Foss Tecator apparatus (Hoganas, Sweden), crude protein by multiplying N-content by 6.25, crude fat by ether extraction (920.39), and crude fiber by the Hennenberg–Stohmann method (978.10) using Fibertec Tecator (Sweden) apparatus. Phosphorus was analyzed after mineralization with nitric acid (HNO_3_) and perchloric acid (HClO_4_) by the ammonium vanadomolybdate method using a spectrophotometer (Specol 11) (Carl Zeiss, Jena) at 470 nm (995.11). Calcium and other mineral (important for the dipeptide synthesis and metabolism) content were determined by the atomic absorption spectrophotometry using AA 240 FS apparatus with SIPS 20 (Varian) (968.08).

### 2.4. Amino Acids Profile

For determination of the amino acids (994.12) samples of used feed components, diets and breast muscles were hydrolyzed with 6 M hydrochloric acid (HCl) for 24 h at 110 °C. Then, amino acids were separated according to Moore and Stein methods [24]. For determination of the sulfur amino acids, the feed samples were oxidized (0 °C, 16 h) with formic acid and hydrogen peroxide (H_2_O_2_) (9:1/*v*:*v*) prior to HCl hydrolysis followed by separation on Analysator AAA 400 Ingos (Prague, Czech Republic) (985.25). For tryptophan content, after alkaline hydrolysis with lithium hydroxide (LiOH) (110 °C, 16 h) and 4-dimethylaminobenzaldehyde (DMAB), the samples were examined spectrometrically at 590 nm according to Landry and Delhaye [25] (988.15). On the basis of the obtained results, the amino acid content in feed mixtures was optimized and further analyzed in complete diets. The energy value of diets was calculated on the basis of data published in the European Tables of Energy Values of Feeds for Poultry [26] and Polish Recommendation of Poultry Feeding [27].

All analyses of the indices of the antioxidative characteristics and amino acid content in muscles were repeated in duplicate.

### 2.5. Radical Scavenging Ability

Antioxidant scavenging activity was measured in the hydrophilic fraction of muscles or blood plasma using the following method: ABTS reducing activity by Re et al. [28]. A decrease in absorbance at wavelength 734 nm was monitored after 10 min of incubation. Radical scavenging activity was expressed as μM Trolox per g wet tissue or mL of plasma; DPPH determination of scavenging activity of 1,1-Diphenyl-2-picrylhydrazyl radical was conducted according to Jang et al. [29]. The absorbance of the DPPH solution was controlled at 517 nm. The ability to scavenge the DPPH radical was calculated from data obtained for standard solutions and expressed as μM Trolox (6-hydroxy-2,5,7,8-tetramethylchroman-2-carboxylic acid) per g wet muscles or mL of plasma; FRAP—ferric reducing antioxidant power analysis was conducted using the spectrophotometric method (593 nm) according to Benzie and Strain [30] (the Fe3+ TPTZ –ferric-tripyridyltriazine). An increase of absorbance at 593 nm was measured after 10 min incubation. FRAP was expressed as μM Fe^2+^ in relation to 1 g wet tissue or mL of plasma.

### 2.6. Activity of Antioxidant Enzymes

Catalase (CAT, EC 1.11.1.6) activity was measured by the Aebi method [31]. A decrease in absorbance was measured at 240 nm after 30 s. One unit (U) of catalase was defined as the amount of blood cell sediment protein (hemolysate) or the amount of tissue needed to decompose 1 mM of H_2_O_2_ per minute.

Superoxide dismutase (SOD, EC 1.15.1.1) activity was analyzed using Cayman’s Superoxide Dismutase Assay Kit No. 7060002. The method is based on the utilization of a tetrazolium salt for the detection of superoxide radicals generated by xanthine oxidase. One unit (U) of SOD activity was defined as the amount of enzyme needed to exhibit 50% dismutation of the superoxide radical and calculated on 1 g of wet tissue or 1 mL of plasma or protein amount in hemolysate.

Glutathione peroxidase (GPx, EC 1.11.1.9) activity was determined spectrophotometrically by measuring the oxidation of nicotinamide adenine dinucleotide phosphate (NADPH) in the presence of reduced glutathione and hydrogen peroxide according to the manufacturer’s instructions (Cayman Chemicals, Kit No 703102). One unit was defined as the amount of enzyme that will cause the oxidation of 1.0 mM of NADPH to NADP+ per min at 25 °C. The activity of the enzyme was calculated on 1 g wet muscles or 1 mL plasma or protein amount in hemolysate.

### 2.7. Histidine Dipeptides Content—HPLC Method

The hydrophilic fractions from muscle tissue were deproteinized [32], which included filtration through glass wool, the addition of 3 vol of methanol (HPLC grade), keeping in the freezer for 30 min, and samples’ final purification by centrifugation (12,000× g) (Sigma, 3K30). Plasma was analyzed after deproteinization by an SPE system (Solid Phase Extraction). The methodology of HPLC analysis was based on Henderson et al. [33]. The sample solution was reacted with O-phthalaldehyde (OPA pre-column derivation method) before being injected into HPLC. Chromatography was performed in an HPLC 1100 Series system (Agilent Technologies) equipped with a fluorescent detector. The separation was monitored at a wavelength of 340/450 nm (excitation/emission) for 0–15 min and a wavelength of 266/305 nm after 15 min. The chromatographic separation was developed using a ZORBAX Eclipse-AAA column (4.6 × 150 mm, 3.5 μm, Agilent Technologies). Mobile phases consisted of solvent A, containing 40 mM Na_2_HPO_4_, pH 7.8 in water and solvent B containing ACN: MeOH: water (45:45:10). The gradient was programmed as follows: 0% B for 1.9 min, linear gradient from 0% to 57% B for 18.1 min, up to 100% B at 18.6 min and stayed at 100% B for four minutes, then up to 26 min at 0% B. L-carnosine and L-anserine (as L-anserine nitrate salt) from Sigma (Sigma-Aldrich Corporation, St. Louis, MO, USA) were used for the method validation. The linear relationship between dipeptide standards of five different concentrations and peak areas was found. Reproducibility of the method (92–104%) was verified after the addition of the standard dipeptides to chicken blood plasma and muscle homogenate extract.

### 2.8. Statistical Analysis

Obtained experimental data were evaluated statistically by one-way ANOVA using StatSoft Statistica^®^ Software (2009). The differences for all parameters were tested according to the following statistical model:Yij = μ +ai + eij
where Yij is the variance associated with parameter a, μ is the overall mean, ai is the treatment effect, and eij the error term. The individual measurements of 238 chickens (during the experiment two chickens died) and body weight of chickens selected for biochemical parameters, blood and meat parameters, or averages for cages—replications, feed consumption, and feed conversion were treated as the experimental units, and differences between treatments means were analyzed for significance (*p* < 0.05) using Tukey’s test. The data are presented as average values and accompanied by the pooled standard error of the mean (SEM).

## 3. Results

### 3.1. Performance Indices of Chickens

Diets supplemented with β-alanine, L-histidine, and carnosine increased (*p* < 0.05) the body weight of chickens at day 28 post-hatch (Table 3). The highest (*p* < 0.05) feed consumption was noticed for chickens treated with the carnosine supplemented diet (Table 3). The lowest feed conversion ratio was calculated for chickens fed diets enriched with β-alanine, whereas the highest FCR was identified for chickens fed the diet supplemented with carnosine. Although feeding chickens with the carnosine supplemented diet led to an increase of the breast muscle weight (*p* < 0.05), the share of these muscles in the chickens’ body was not affected.

### 3.2. Antioxidant System in Chicken Muscles

Lower (*p* < 0.05) glutathione peroxidase activity compared to other experimental treatments was analyzed in breast muscles collected from chickens fed with the diet containing carnosine (Table 4). β-alanine supplementation of the chicken diet resulted in the highest level of catalase activity.

Total antioxidant capacity of chicken breast muscles measured by FRAP, ABTS, and DPPH radical scavenging methods was not affected by feeding modifications (*p* > 0.05).

Carnosine and L-histidine supplementation in chicken diets affected carnosine (*p* < 0.05) level, as well as the sum of the histidine dipeptides (carnosine and anserine) in breast muscles (Table 4). The highest amount of analyzed carnosine, of about 50% more than in the control, was found in chickens fed diets with added L-histidine. However, the addition of β-alanine to histidine-enriched diet lowered the anserine content in breast muscles, and the amount of both dipeptides after feeding with L-histidine/β-alanine mixture was higher than in the control (less than 20%).

### 3.3. Antioxidant System in Chicken Blood

L-histidine addition to the chicken diets increased (*p* < 0.05) SOD and GPx activities in blood cell sediment. Feeding chickens the diet supplemented with β-alanine resulted in lower SOD and GPx activities in blood cell sediment compared to chickens fed the diet with L-histidine. Both enzyme and free radical scavenging activities of chicken blood plasma were not affected by the applied feeding modifications (Table 5).

The β-alanine-supplemented diet decreased (*p* < 0.05) the concentration of carnosine, as well as the sum of histidine dipeptides, in chicken blood plasma (Table 5). The addition of L-histidine and carnosine to the diet resulted in higher carnosine and the sum of histidine dipeptides in blood plasma compared to the β-alanine supplemented diet.

### 3.4. Amino Acids Content in Chicken Muscles

Breast muscles of chickens fed diets supplemented with β-alanine and L-histidine or both were characterized by lower threonine and asparagine content in meat than the control group (Table 6). Carnosine addition to the diet resulted in higher (*p* < 0.05) asparagine and lower threonine content. Diet supplementation with β-alanine or carnosine resulted in a higher content of β-alanine and tryptophan in chicken breast muscles.

## 4. Discussion

Growth and feed conversion ratio indices of chickens were positively affected by diet supplementation with β-alanine and L-histidine or carnosine. Although the requirement of histidine for young chickens amounts to 3.5 g/kg, the content in the control diet was 5.8 g/kg. In contrast, the supplementation of histidine and endogenous amino acid β-alanine was effective. The effect of β-alanine supplementation can show that this amino acid might be essential for fast-growing chickens, optimally for FCR at 1.1 g/kg [2], which indicated an increase in the final whole weight gain even after 5.0 g/kg carnosine was supplemented in the diet [5]. Moreover, cited authors found a significant increase in breast muscle weight confirmed also by the presented investigations.

The possible reason for an increase in live animal weight as well as breast muscle weight in the studies could be the stimulating effect of histidine or carnosine on insulin secretion reported by Leblanc and Saucy [34] and Yamano et al. [35], which can result in more intensive growth. Although that effect can also be negative because histidine addition or administration at a level higher than 10–15 g/kg suppressed the growth rate, probably due to feed consumption inhibition connected with high histamine concentration [36].

β-alanine positively affected final BW but not breast muscle weight. It seems that the role of β-alanine in growth performance should be directed to more complex action than in the case of histidine or its derivatives. β-alanine plays various roles in organisms as a neurotransmitter and protects many systems from toxic agents, like β-amyloid in the brain or CCl4-induced liver injuries [37]. This antitoxic effect may be responsible for better growth. Another effect of β-alanine supplementation can be the usage of this β amino acid for pantothenate synthesis, which is, in a limited range, conducted by gut microflora [38]. Because pantothenate is a growth promoter [39], its higher level could explain the increase in BW and also the lower feed conversion factor, as observed in the present experiment. Furthermore, Jacob et al. [40] showed improved feed gain ratio as the effect of feeding chickens with a diet containing β-alanine.

Supplementation of chicken diets with histidine, its mixture with β-alanine, or histidine-containing dipeptides increased mainly carnosine by about 38–50% and, to a smaller degree, the total sum of dipeptides in breast muscles due to a quite stable anserine content. It is worth underlining that histidine supplementation was only about 30% higher than in the control diet. Collected data correspond to the increase in carnosine by about 60% in chicken breast muscles observed by Haug et al. [17], even at lower histidine supplementation (0.1%) in the feed, which was also rich in histidine components. Robbins et al. [3] observed 2–3 times higher carnosine content in breast muscles fed a diet containing about 0.3% L-histidine, which is the limit value for deficiency of this amino acid in relation to feeding with low content or without histidine. In general, all these facts enabled the formulation of the hypothesis that histidine is a critical factor for carnosine synthesis in chicken muscles but it does not concern anserine synthesis. The small increase in anserine observed in the study was similar to the results [4,17,41], which stated only a slight increase or even a decrease in anserine as a result of histidine or β-alanine diet supplementation. This could suggest that histidine is used for carnosine synthesis but diet supplementation has no effect on the process of the histidine methylation creating component of anserine. There are two possible ways of anserine synthesis: via carnosine methylation [42] or direct synthesis from methylhistidine and β-alanine. The latter is catalyzed, like in the case of carnosine, by synthetase which is able to use histidine derivatives as a substrate also [43].

A quite high diet supplementation with β-alanine in relation to histidine (3 times excess if expressed in mM) resulted only in a tendency to increase carnosine concentration in chicken muscles, and the effect was lower than histidine addition. When both amino acids were supplemented in the diet, carnosine concentration in muscles was similar to the histidine fed group. The results are contradictory to observations made for humans and other mammals, where β-alanine supplementation led to the increase of carnosine as the effect of activation of muscle carnosine synthetase [44].

The content of histidine dipeptides in blood plasma is quite low because of the action of serum carnosinases [45,46]. Such relation was also observed in the present study, where only a slight increase in carnosine after dipeptide supplementation was observed. In addition to that, an unexpected decrease in plasma carnosine of chickens supplemented with β-alanine was found.

Possible explanations for this relationship are the many other physiological functions played by β-alanine, like being a neurotransmitter [38], which is used by the nervous system. Accelerated turnover of carnosine in blood as the effect of β-alanine supplementation can result from the activation of chicken erythrocyte synthetase described by Seely and Marshall [47]. Plasma anserine determined at a lower concentration than carnosine was not affected by amino acids incorporated into the diet. It was probably caused by the lower activity of plasma carnosinase CN1 in relation to anserine, which is degraded 200 times slower than carnosine. In addition to that, anserine can inhibit carnosinase activity [46,48]. Such characteristics can lead to achieving a stable anserine level.

Low concentration of both carnosine and anserine in plasma, despite some fluctuations, has no effect on total antioxidant capacity. Only a tendency to increase FRAP as the effect of amino acid supplementation was observed. It corresponds to the observations of Hu et al. [5] concerning an increase of ferric ion reduction ability in chicken plasma but at about double the carnosine supplementation than in these studies. The tendency to increase FRAP in plasma is probably caused by an increase of free amino acid content, i.e., histidine and β-alanine, resulting also from partial hydrolysis of carnosine. The β-alanine has weaker antioxidant capacity compared to histidine [49] but, at an extensive level of β-alanine supplementation, the content of free β-alanine can be quite high. Eventual high level of free amino acids did not affect enzymatic antioxidant systems (SOD, GPx) and radical scavenging activity (ABTS, DPPH) in plasma of chickens fed diets with the addition of histidine, β-alanine, or carnosine.

Erythrocytes are susceptible to oxidative stress due to high polyunsaturated fatty acid content [50]. The main protection mechanism of erythrocytes from oxidation is the enzymatic system (i.e., triad SOD, GPx, CAT), which was strongly activated in the present study by histidine supplementation. The protection ability of the supplements in relation to SOD can be shown as the following sequence: Histidine > Carnosine > Histidine + β-alanine. The highest ability of SOD protection after histidine supplementation can be the result of the antioxidant ability of free amino acids to inactivate prooxidants and possible protection of the histidine residue near the reactive center of the SOD molecule, which is preferably oxidized by peroxides leading to enzyme inactivation [51,52]. The better protection of SOD by histidine than carnosine supplementation observed can be caused by a higher ability of histidine than dipeptide to achieve SOD from inactivation as a result of glycation with carbohydrates [53].

The noted strong effect of histidine supplementation on GPx content in blood cell sediment (mainly erythrocytes) is more difficult to explain than in the case of SOD. However, lower malonaldehyde content and increased erythrocyte GPx and catalase as the effect of feeding rats with histidine and carnosine supplementation were shown [36]. The lack of the effect of carnosine on GPx, observed in the studies, could be the result of the higher capacity of carnosine than histidine to create complexes with metals such as zinc, among others [54]. Some free ions, like Zn, can indirectly affect metalloselenite-activating GPx [55].

Catalase activation by histidine and carnosine supplementation can be explained by the same mechanism as in the case of SOD. Histidine or carnosine is able to inactivate Cu^2+^ ion catalyzing oxidation of amino acid residues in the reactive center of enzymes [56]. Catalase increase was not observed when the diet was supplemented with β-alanine, also together with histidine. Supplementation with β-alanine caused a reduction in CAT activity. The reason for lower activity of antioxidants including erythrocytes after β-alanine supplementation in comparison to histidine or carnosine treatments could be the higher susceptibility of β-alanine to Fenton-type oxidation, which has to be one of the main sources of peroxides in a system with high hemoglobin content [49]. In chicken muscles, as opposed to blood cells (sediment rich in erythrocytes), catalase activity increased after β-alanine supplementation. Such a difference in CAT activity between erythrocytes (low) and other organs such as the lung (high) was also found by Benzer and Yilmaz [57] as the effect of oxidation stress caused by infection. Most often at the deficiency of antioxidants or other stress conditions, increased muscle catalase was observed, while SOD and GPx were stable or decreased [58]. So, the reason for an increase in CAT when diet with β-alanine was supplemented could be lower levels of cytosolic antioxidants (i.e., dipeptides) in the present study.

The decrease in GPx after carnosine supplementation observed in the studies corresponds with the results of other authors [59] who found lowered GPx after diet supplementation with antioxidants (α-tocopherol). Ma et al. [60] also found no effect of carnosine supplementation (0.1%) on GPx activity in pig muscles, while other antioxidant enzymes were affected. The newest publications showed that antioxidative enzymes, such as SOD and GPx, can express higher activity in thigh muscles of chickens supplemented with carnosine [11].

The antioxidant effect of L-carnosine has already been proved and the relation between its content and shelf-life of fresh meat has been established [61]. Anyway, no relation between diet and ABTS, DPPH, or FRAP scavenging ability in muscles at varying carnosine content in the present study was documented. The ability of carnosine, imidazole ring, histidine, and other amino acids (β-alanine) to scavenge free radicals is still under discussion [62,63]. The ability of histidine or carnosine to chelate prooxidative ions, like Cu or Ni, scavenge carbonyl and nitrogen reactive species [64,65], inhibit protein carbonylation and glycoxidation [66], as well as the antiglycation action of carnosine is fully documented, but in experiments concerning the ability to scavenge radicals like ABTS or damage to reactive oxygen species (ROS), contradictory results have been obtained [1,18,62]. It could be the reason for no observed effects of diet supplementation with amino acids on the total antioxidant capacity of muscles expressed as the ability to scavenge model-free radicals.

Despite small changes in the amino acid shares, some fluctuations indicate different muscle metabolism depending on dietary supplementation. It is demonstrated among others by the lowering of branched-chain amino acid content after the addition of histidine, β-alanine, or carnosine to the diet. Because the amount of amino acids was expressed in relation to crude protein, the observed changes can be due to the increased content of muscle histidine and β-alanine as the effect of diet supplementation. There is no linear relationship between the content of both amino acid and dipeptide levels in muscles. The probable reason for such relation is the balance between free and bound dipeptide amino acids, which was also stated by Haug et al. [17].

## 5. Conclusions

It can be concluded that the addition of L-histidine and carnosine to diets resulted in positive effects on chicken performance and quality factors, but supplementation of diets with L-histidine is more cost-effective. Histidine supplementation was a critical factor for increased carnosine content in chicken breast muscles, which was shifted by 30% but did not affect anserine levels. Moreover, the diet enriched with carnosine also increased histidine dipeptide content in muscles, while the effect of the addition of β-alanine was low. L-His and carnosine in the diet activated the enzymatic antioxidant system of chicken blood.

## Figures and Tables

**Table 1 antioxidants-09-01093-t001:** Composition of basic control diet.

Ingredient	Amount
Wheat [g/kg]	396.00
Maize [g/kg]	200.00
Rapeseed oil [g/kg]	38.00
Soyabean meal [g/kg]	327.00
Dicalcium phosphate [g/kg]	19.65
Limestone [g/kg]	2.45
NaCl [g/kg]	3.50
Premix dka-s ^a^ [g/kg]	10.00
DL—Met 98% [g/kg]	2.32
L—Lys 98% [g/kg]	1.03
Metabolic energy ^b^ [MJ/kg]	
Chemical analysis	11.98
Crude protein [g/kg]	215.90
Crude fiber [g/kg]	32.30
NDF [g/kg]	11.56
ADF [g/kg]	3.99
Crude fat [g/kg]	56.90
Crude ash [g/kg]	65.20
Ca [g/kg]	9.17
P_available_ [g/kg]	4.30
Na [g/kg]	1.60
Mg [g/kg]	1.62
Cu [mg/kg]	8.93
Mn [mg/kg]	80.00
Fe [mg/kg]	365.65
Zn [mg/kg]	59.60

^a^ In 1 kg of premix: vitamin A (retinyl acetate) 10,000 IU, vitamin D_3_ (cholecalciferol) 2000 IU, vitamin E (DL—α—tocopheryl acetate) 20 mg, vitamin K (hetrazeen vit.K_3_ free from menadione) 3 mg, vitamin B_1_ (thiamine mononitrate) 2.5 mg, vitamin B_6_ (pyridoxine HCl) 0.4 mg, vitamin B_12_ (cyanocobalamin) 0.015 mg, nicotic acid 25 mg, pantothenic acid 8 mg, folic acid 1.2 mg, choline chloride 450 mg, DL—methionine 1.0 mg, Mn (as MnO_2_) 74 mg, Fe (as FeSO_4_⋅H_2_O) 30 mg, Zn (as ZnO) 45 mg, Cu (as CuSO_4_⋅5 H_2_O) 4 mg, Co (as CoCO_3_) 0.4 mg, J (as Ca(JO_3_)_2_) 0.3 mg. ^b^ Energy value of diets was calculated with data of European Tables of Energy Values of Feed for Poultry (1989).

**Table 2 antioxidants-09-01093-t002:** Amino acid content in diets.

Amino Acid	Treatments [g/kg]				
	C	ExpH	ExpA	ExpH+A	ExpCar
Histidine	-	1.8 (12)	-	1.8 (12)	-
β-Alanine	-	0	3.2 (36)	3.2 (36)	-
Carnosine	-	-	-	-	2.7 (12) eq. of 1.8 (12) His

C—control diet; ExpH—diet with added L—histidine; ExpA—diet with added β-alanine, ExpH+A—diet with added L-histidine and β-alanine; ExpCar—diet with added carnosine.

**Table 3 antioxidants-09-01093-t003:** Production results of chickens at day 28 post-hatch.

Item	Treatments [g/kg]					SEM ^d^	*p*-Value
	C	ExpH	ExpA	ExpH+A	ExpCar		
Body weight [kg]	1.19 ^a^	1.23 ^b^	1.31 ^b^	1.28 ^b^	1.28 ^b^	0.012	0.022
Feed consumption [kg]	1.76 ^a^	1.91 ^a,b^	1.86 ^a,b^	1.86 ^a,b^	2.12 ^b^	0.037	0.016
Feed conversion ratio [kg/kg BW]	1.63 ^a^	1.55 ^a,b^	1.41 ^b^	1.45 ^a,b^	1.65 ^a^	0.028	0.009
Weight of breast muscles without skin [g]	173.8 ^a^	198.3 ^a,b^	196.6 ^a,b^	195.5 ^a,b^	207.7 ^b^	3.56	0.041
Share of breast muscles in BW [%]	14.59 ^a^	16.13 ^b^	15.01 ^a,b^	15.27 ^a,b^	16.17 ^b^	0.186	0.046

Differences in rows signed with a, b significant by *p* < 0.05; *n* = 238. ^c^ The body weight of 1-day chickens was 43 ± 2 g. ^d^ SEM, standard error of the mean. C—control diet; ExpH—diet with added L-histidine; ExpA—diet with added β-alanine, ExpH+A—diet with added L-histidine and β-alanine; ExpCar—diet with added carnosine.

**Table 4 antioxidants-09-01093-t004:** Antioxidant characteristics of chicken breast muscle.

Item	Treatments [g/kg]					SEM ^d^	*p*-Value
	C	ExpH	ExpA	ExpH+A	ExpCar		
SOD [U/g]	2.92	3.29	3.16	2.67	2.53	0.246	0.449
GPx [U/g]	0.75 ^a,b^	0.77 ^b^	0.76 ^b^	0.76 ^b^	0.64 ^a^	0.032	0.008
CAT [U/g]	296 ^a^	277 ^a^	398 ^b^	291 ^a^	294 ^a^	24.80	0.003
ABST [mM Trolox/g]	2.37	2.38	2.40	2.45	2.47	0.315	0.490
DPPH [mM Trolox/g]	13.5	12.6	11.7	12.5	11.9	1.380	0.062
FRAP [mM Fe/g]	0.17	0.16	0.15	0.16	0.17	0.017	0.384
Carnosine [mg/g tissue]	1.22 ^a^	1.83 ^b^	1.51 ^a,b^	1.80 ^b^	1.68 ^b^	0.127	0.003
Anserine [mg/g tissue]	2.37 ^a,b^	2.46 ^b^	2.36 ^a,b^	2.25 ^a^	2.47 ^b^	0.047	0.012
Sum of dipeptides [mg/g tissue]	3.60 ^a^	4.30 ^c^	3.88 ^a,b^	4.05 ^b,c^	4.16 ^b,c^	0.129	0.001

Differences in rows signed with a, b, c significant by *p* < 0.05; *n* = 60. ^d^ SEM, standard error of the mean. C—control diet; ExpH—diet with added L-histidine; ExpA—diet with added β-alanine, ExpH+A—diet with added L-histidine and β-alanine; ExpCar—diet with added carnosine.

**Table 5 antioxidants-09-01093-t005:** Antioxidant enzymes activities in blood cells sediment as well as in blood plasma and selected antioxidant content in blood plasma.

Item	Treatments, g/kg					SEM ^d^	*p*-Value
	C	ExpH	ExpA	ExpH+A	ExpCar		
Blood cells sediment							
CAT, U/mg	272 ^a,b^	349 ^b^	169 ^a^	185 ^a^	367 ^b^	47.60	0.001
SOD, U/mg	0.156 ^a^	0.276 ^b^	0.151 ^a^	0.196 ^a,b^	0.228 ^a,b^	0.025	0.023
GPx, U/mg	0.413 ^a^	0.516 ^b^	0.360 ^a^	0.359 ^a^	0.357 ^a^	0.026	0.006
Blood plasma							
SOD [U/mL]	2.34	1.97	2.45	1.98	2.15	0.171	0.173
GPx [U/mL]	1.25	1.16	1.34	1.28	1.37	0.100	0.307
ABTS [mM Trolox/mL]	2.76	2.78	2.70	2.77	2.75	0.359	0.391
DPPH [mM Trolox/mL]	10.5	10.1	10.4	10.4	9.9	0.490	0.103
FRAP [mM Fe/mL]	0.143	0.159	0.150	0.156	0.153	0.004	0.080
Carnosine [µg/g]	143 ^b,c^	148 ^b,c^	89 ^a^	122 ^a,b^	167 ^c^	14.2	0.001
Anserine [µg/g]	25.1	27.4	22.2	23.1	19.7	2.40	0.421
Sum of dipeptides [µg/g]	168 ^b^	175 ^b^	111 ^a^	145 ^a,b^	187 ^b^	15.1	0.002

Differences in rows signed with a, b, c significant by *p* < 0.05; *n* = 60. ^d^ SEM, standard error of the mean, C—control diet; ExpH—diet with added L-histidine; ExpA—diet with added β-alanine, ExpH+A—diet with added L-histidine and β-alanine; ExpCar—diet with added carnosine.

**Table 6 antioxidants-09-01093-t006:** Content of amino acids in breast muscles of broiler chickens, g/100 g of crude protein.

Amino Acid	Treatments, g/kg					SEM ^d^	*p*-Value
	C	ExpH	ExpA	ExpH+A	ExpCar		
Asp	9.34 ^b^	9.08 ^a^	9.08 ^a^	9.02 ^a^	9.48 ^c^	0.082	0.001
Thr	4.76 ^c^	4.63 ^b^	4.59 ^b^	4.55 ^b^	4.45 ^a^	0.050	0.001
Ser	4.12	4.05	4.03	4.01	3.89	0.044	0.001
Glu	14.5	14.3	14.4	13.9	13.7	0.167	0.152
Pro	3.34	3.35	3.30	3.23	3.98	0.033	0.259
Gly	4.55	4.30	4.28	4.48	4.10	0.076	0.335
Ala	5.78	5.58	5.57	5.48	5.52	0.051	0.445
Val	4.89	4.65	4.53	4.36	4.57	0.050	0.107
Ile	4.42	4.16	4.12	3.98	4.28	0.035	0.058
Leu	7.93	7.69	7.70	7.54	7.63	0.069	0.329
Tyr	3.13	3.07	3.04	3.10	3.17	0.023	0.362
Phe	3.82	3.68	3.72	3.81	3.98	0.024	0.056
His	5.11	5.59	5.59	5.76	5.71	0.071	0.248
β—Ala	0.49 ^a^	0.47 ^a^	0.58 ^b^	0.58 ^b^	0.54 ^b^	0.008	0.001
Lys	9.30	8.87	8.96	8.82	8.84	0.087	0.334
Arg	7.42	7.08	7.27	7.35	7.41	0.058	0.150
Cys	1.06	1.05	1.01	1.01	1.15	0.006	0.337
Met	2.81	2.70	2.65	2.67	3.00	0.014	0.275
Trp	1.09 ^c^	1.11 ^c^	0.99 ^a^	1.00 ^a^	1.04 ^b^	0.015	0.001

Differences in rows signed with a, b, c significant by *p* < 0.05; *n* = 60. ^d^ SEM, standard error of the mean. C—control diet; ExpH—diet with added L-histidine; ExpA—diet with added β-alanine, ExpH+A—diet with added L-histidine and β-alanine; ExpCar—diet with added carnosine.

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
