# Peer review of "Antioxidative Characteristics of Chicken Breast Meat and Blood after Diet Supplementation with Carnosine, L-histidine, and β-alanine"

_antioxidants, 2020, doi:10.3390/antiox9111093_

Round 1

Reviewer 1 Report

Dear Authors,

Please see my comments:

1- Please provide P-valuse (only significant ones) and some obtained data in the abstract. 

2-Line 36-40. Line format is incorrect, please fix it. 

3- line 67-69. assigned to five treatments: (i) control 

To avoid confusion for readers, for other diets add control as well. For example: Control + L-histidine (ExpH). Please do the same for all the diets.

4- Line 70. Cages size?

5- Line 90. I would recommend you to cite the reference. 

6-All tables titles should be in center. Please check the journal's format. 

7- Line 96. what is the point of individual weight when the chickens were not tagged? 

8- Line 100. were slaughtered after stunning with cervical dislocation. This statement is confusing, stunning with cervical dislocation, each one is a different method. If you did stunning, provide manufacture information, if you did dislocation provide more information how it was done.

9-Line 105. collecting blood the way it was done might affect all your results, the collect blood may not be pure. Best way is to collect from wing vein or neck before killing birds. Could you please explain why the authors did not consider this before starting the experiment? 

10- Why 28 days only, mostly this time is 5-6 weeks?

11- Table 4. Superscripts for Gpx looks incorrect. P-values are significant but Superscripts show something else. Please be consistent, some have 1 or 2 or 3. maybe have 2 for all, and 3 if its necessary. 

12- Conclusion section. The first paragraph can be combined with second paragraph. Some statements are like introduction. Please only provide significant findings of your study. 

Regards,  

Author Response

Dear Reviewer 1,

Thank you for your reviewing our manuscript. All comments and questions are answered in the attached file.

Best regards,

Malgorzata

Reviewer 2 Report

In general, I find the presented study interesting. The whole presentation of the study is also of high quality.

However, I have a few questions/remarks that needs clarifications from the Authors, especially regarding M&M section:

  • First of all, please rewrite the description of the experimental groups. It's hard to follow, and at least for me it is not compatible with the Table 2. Where is the addition of Carnosine in this table? Furthermore, please explain on what basis did the Authors choose the doses. Additionally, please add the information about the amino acids that you used (producer etc...).
  • My main concern is that the experiment is not completely randomized. Please explain why in the group with carnosine (ExpCar) did you apply higher doses of His as well (Table 2)? At least one additional treatment (when only carnosine would be used, and the level of histidine should be the same as in the control group) would be beneficial for the whole study and your conclusions.
  • The experiment lasted 28 days - only one diet was used for the whole period? There was no differentiation to starter/grower diet?
  • Please explain, when was the study carried out. Because I have noticed that the Local Ethical Commission approved it in 2009?
  • Also, please explain, why did the Authors decide to perform 3 different methods of radical scavenging ability assessments? Did you deproteinise the blood plasma before DPPH assessment?
  • How the Authors calculated the activity of antioxidant enzymes in erythrocytes and expressed it as "U/mg of erythrocytes"??? Please explain.
  • The experimental model should be expanded by performing contrast analysis, which could provide a better understanding of the data.
  • Please check, Lines 229-231/ Table 4: "The highest amount of analyzed dipeptides, of about 50% more than in control, was found in chicken fed diets with added L-histidine. " It is true when you consider carnosine, but not when you check the sum of dipeptides.
  • Table 5: How would you explain the possible mechanism why the addition of histidine alone (ExpH group) has such high differences between other groups that also have received higher amount of histidine (ExpH+A i ExpCar)?
  • In discussion section, the Authors support their assessments on the study on invertebrates [eg. 31], I would suggest removing it and replacing with other study.
  • Please once again check the manuscript, as there are a few linguistic mistakes, for instance:
    • Line 37: ..considered to be an immunomodulator, for tissue repairing and hypoglycemic effect in several species including humans."
    • Line 75: "Control diet was supplemented with L–histidine on the level exceed 30% of control diet,..."
    • Line 80: " Used vitamins/minerals premix was free of feed supplements,... "
    • Line 401-402: "Diet enriched with carnosine, which improved both chickens performance and histidine dipeptides content in muscles. "

Author Response

Dear Reviewer,

Thank you for your reviewing our manuscript. All comments and questions are answered in the attached file.

Best regards,

Malgorzata

Reviewer 3 Report

The present work was good. The quality of the English is also a concern.

Your manuscript would greatly improve from editing by a native English speaker.

Re-write the abstract. Suggested to remove the subheading in abstract with numbers. Abbreviations must be defined at first mention and used consistently thereafter. As well in manuscripts [Eg. Line no. 45, 262].

Background of the aim, not strong enough.

In materials and methods, - Check the subheading style according to journal format.

Line no. 67-68: There were no details regarding treatments. Must clearly mention the supplement levels.

Line no.: HNo3 and HCL04 – numeric should be in subscripts

Discussion section should be more improve, suggested to proof read before submission.

Suggested to remove the words – “own study or own experiments”. Not fit with scientifically.  

Line no. 347 - metals a. o. zinc? What meaning for a. o? Also in Line no. 382

Line no. 356 - Stadtman and Berlett, 1991 – cite the desired number.

In tables: Explains the abbreviations in footnote.

Reference Section – follow the Journal style.

Author Response

(The authors gave the same response as above.)

Round 2

Reviewer 1 Report

Dear Authors, 

Thank you for the revised version. I have no more questions regarding the manuscript. 

Regards, 

Author Response

Dear Reviewer 1,

Thank you very much. All authors appreciate your hard work and valuable comments.

Best regards,

Malgorzata

Reviewer 2 Report

Dear Authors,

Thank you for addressing my remarks and questions. I have one more question, though. Can you please check once again the methodology for the assessment of enzymes activity in erythrocyte lysates? When comes to the analysis of enzymes activity in erythrocytes most methods analyse the enzyme activity as U/mg of protein or U/mg haemoglobin. 

I have not seen activity of enzymes expressed as "U per mg of blood cell sediment", especially that the Authors have mentioned in the M&M (sub-section 2.6) that the activity was estimated per mg of protein e.g. "One unit (U) of catalase was defined as the amount of erythrocyte blood cells sediment protein (hemolysate) or amount of tissue needed to decompose 1 186 mM of H2O2 per minute." 

Now, if it was expressed per mg of protein, please add information what method did the Authors use for protein determination.

Please check again, because this can be confusing and misleading for readers and the methodology should be described clearly so that it can be reproduced.

Kind regards

Author Response

Dear Reviewer 2,

Thank you very much for your valuable comments. 

We apologize for this lack of information in our manuscript. This happened while the manuscript was prepared and we have not notice it.

For the determination of enzymatic activity we used Cayman tests and (as we now included in the section M&M 2.2 Blood collection and muscles preparation for analyses), the Biuret method (Gornall et al., 1949) for protein determination in order to be able to express the enzyme activity in U/mg protein. 

Thank you once more for your hard work and comments, that contributed to our manuscript and make it better.

Best regards,

Malgorzata